# Impact of having a child on physical activity in the UK: a scoping review protocol

Matthew Northcote ,[1] Charlie Foster,[1] Richard Pulsford,[2] Fiona Spotswood[3]

¹School for Policy Studies, University of Bristol, Bristol, UK
²College of Life and Environmental Sciences, University of Exeter, Exeter, UK
³School of Management, University of Bristol, Bristol, UK

**Correspondence to**
Matthew Northcote;
av20205@bristol.ac.uk

## ABSTRACT

**Introduction** Throughout the life course, there are major life transitions that are associated with reduced physical activity, which may have further implications for health and well-being. Having a child is one such transition that has been identified as a critical transformative experience and window for intervention. We will conduct a scoping review of available evidence exploring the impact of having a child on physical activity in the UK.

**Methods and analysis** We will use best-practice methodological frameworks to map key concepts and available evidence, summarise and disseminate findings to stakeholders, and identify knowledge gaps. A three-step search strategy will identify primary research studies, including reviews, from published and grey literature, exploring the impact of having a child on physical activity in the UK, from the preconception period, throughout pregnancy, the postpartum period, and into parenthood. An initial limited search will identify relevant reviews, from which keywords and index terms will be extracted. We will conduct searches of CINAHL, Embase, Medline, PsycINFO and Web of Science to identify relevant articles written in English from inception to February 2022. Two reviewers will independently screen titles and abstracts of identified studies for inclusion and chart data, with a third reviewer resolving any conflicts. Backwards citation tracking will identify any additional studies. We will conduct numerical and thematic analysis to map data in tabular and diagrammatic format and provide a description of findings by theme.

**Ethics and dissemination** Ethical approval is not required for this scoping review. We will disseminate findings to stakeholders through publications, conferences, social media platforms and in-person communications. Consultations with key stakeholders, with their unique expertise and perspectives, will provide greater insight. We will establish the main priorities for future research to inform the research questions of subsequent studies.

**Scoping review registration** Open Science Framework (https://osf.io/gtqa4/) DOI 10.17605/OSF.IO/GTQA4.

## STRENGTHS AND LIMITATIONS OF THIS STUDY

⇒ We will use a scoping review methodology to provide an overview of available evidence exploring the impact of having a child on physical activity in the UK.

⇒ Scoping reviews are a form of literature review which will allow us to map key concepts and available evidence, exploring the extent, range and nature of existing evidence.

⇒ These methods will allow us to use broad research questions to define and clarify existing concepts, summarise and disseminate findings to stakeholders, identify knowledge gaps and determine the main priorities for future research.

⇒ We will use best-practice methodological frameworks to guide the conduct of this scoping review and established guidelines to inform reporting.

⇒ It is important to note that scoping reviews do not address more focused research questions, like in systematic reviews, nor do they generally include formal critical appraisal of individual sources of evidence using tools.

## INTRODUCTION

Physical activity (PA) can be defined as 'any bodily movement produced by skeletal muscles that results in energy expenditure'.[1] The benefits of regular PA participation as a key component of physical, mental, and social health and well-being has been well established.[2] For example, PA can reduce the risk of disease, manage existing conditions, and develop and maintain physical, mental, and social function.[3]

Recent evidence from national health surveys in the UK indicates that a large proportion of adults (aged 16 years and over) are less physically active than the current UK Chief Medical Officer's PA guidelines recommend.[3] Specifically, 34% in England in 2016[4]; 45% in Northern Ireland between 2016 and 2017[5]; 54% in Scotland in 2020[6]; and 44% in Wales between 2020 and 2021[7] did not meet PA guidelines for aerobic activity (ie, at least 150 min of moderate-intensity activity, or 75 min of vigorous-intensity activity, or shorter durations of very vigorous-intensity activity, or combination of moderate-intensity, vigorous-intensity, and very vigorous-intensity activity each week). It is necessary here to note the likely negative impact of lockdown measures to curb the spread of the 2019 coronavirus in surveys conducted between 2020 and

2021. Moreover, of those who met recommendations for aerobic activity, over one-third, 35% in England in 2016[4] and 37% in Scotland in 2019,[8] did not meet that for muscle-strengthening activity (ie, at least 2 or more days each week).

According to PA guidelines, adults are to minimise the amount of time spent being sedentary and break up long periods of inactivity with at least light-intensity activity.[3] Sedentary behaviour (SB) can be defined as 'any waking behaviour characterised by an energy expenditure ≤1.5 (metabolic equivalents of task (METs)) while in a sitting or reclining posture'.[9] In adults, high volumes of SB heighten the risk of all-cause mortality, cardiovascular mortality, cardiovascular disease incidence and type 2 diabetes incidence.[2 10–12] In England, men are more likely than women to have increased time spent being sedentary during both the weekday (4.9 hours and 4.7 hours, respectively) and weekend (5.4 hours and 5.1 hours, respectively).[13]

Although physical inactivity and SB are distinct constructs,[14] PA, including light-intensity (ie, 1.6–2.9 METs), moderate-intensity (ie, 3–5.9 METs) and vigorous-intensity (≥6 METs) activities, and SB exist on the same energy expenditure spectrum.[15] In fact, if we assume adults sleep an average of 8 hours each night, only around 2% of our waking time is required to meet PA guidelines for aerobic activity in the UK.[3] Meanwhile, the remaining 98% of our waking time is spent in SB and light-intensity activity, including static (eg, standing) and ambulatory activities.[14] As a result, it is important to explore activities across the energy expenditure spectrum, not merely the least practised moderate-to-vigorous-intensity PA (MVPA). Furthermore, PA participation modifies the effect of SB on mortality, where higher amounts of MVPA can mitigate all-cause, cardiovascular disease and cancer-related deaths.[2 10 16] For example, high levels of moderate-intensity activity (ie, 60– 75 min per day) has been found to eliminate the risk of mortality associated with high sitting time, while only attenuating the risk associated with high television viewing time.[16]

In the UK, physical inactivity accounts for one in six deaths and has an estimated annual cost of £7.4 billion, including £0.9 billion to the National Health Service (NHS) alone.[17] Moreover, prolonged SB (≥6 hours each day) was estimated to cost the NHS £0.8 billion between 2016 and 2017, which, if eliminated, could have avoided 69 276 deaths.[18] Considering the prevalence and economic and social burden of physical inactivity and SB, it is important to identify populations that are less physically active and/or more sedentary and characterise and understand these contexts to develop targeted interventions which promote PA.

Throughout the life course, there are major life events and transitions that are associated with reduced PA, which may have further implications for health and well-being.[19] Life events 'mark the beginning or end of a specific status. A status is a nominal variable with at least two levels'.[20] For example, the life event of parenthood, or having a child, could be conceptualised to involve multiple consecutive shifts in status from preconception (or non-parent) to pregnant, post partum (until 6 months after giving birth[21]), then finally into parenthood (or parent). Notably, life events can bring about periods of biological, psychological, and/or social adaption and readjustment which can subsequently lead to behaviour change, such as reduced PA participation.[22] Life transitions are conceptualised through these adaptive periods and reflect the processual character and temporality of life events, including their anticipation (to the extent that they can be anticipated), history and eventual aftermath.[22] As a result, both life events and transitions are inherently interconnected yet differ in their temporal characteristics. For instance, life events are singular occurrences which mark a change in status, while life transitions represent the duration of that given status change.[20 22] These conceptualisations of life events and transitions exclude minor life events, such as daily hassles; slow transitions without clearly identifiable life events, like ageing; and non-events, such as not having a child.[20 22]

Parenthood, or having a child, is one such life event and transition that has been identified as a critical transformative experience and window for adult health and well-being and intervention.[23] Moreover, having a child is often interconnected with other life events and transitions, including changes to relationship status, and living situation, which may further compound these effects.[24] As a result, the impact of having a child on PA should be characterised and understood in its entirety, from the preconception period, throughout pregnancy, the postpartum period, and beyond into parenthood, and contextually among other related life events and transitions.

The latest reviews of existing evidence exploring the impact of having a child on PA by Bellows-Riecken and Rhodes[25] and Corder *et al*[26] suggest that the transition into parenthood, becoming a parent, generally reduces PA participation, particularly in new mothers. Although both reviews provide valuable characterisations and understandings of PA during this time, their scope focuses on the differences between parents and non-parents and does not consider the other relevant changes in status embedded in the 'having' of a child, such as becoming pregnant and entering the postpartum period, with their inherent temporal and processual characteristics. In addition, although both reviews include quantitative and qualitative evidence, these were only sourced from published literature, consequently missing the potentially rich insights afforded by grey literature. Both reviews also chose to forgo the characterisation and understanding of the relationship between PA and SB. What is more, these reviews included international sources of evidence from a variety of countries, namely the USA, Australia, Canada and the UK, among others from Europe. While these findings may be generalisable to other western countries, it is important to ensure findings are culturally and contextually relevant to the UK, making them more applicable and practicable within this setting. Given that there were

585 195 live births in England in 2020,[27] it is important to provide an overview of available evidence exploring the impact of having a child in the UK.

Recent evidence exploring the impact of having a child on PA by intensity, including SB, indicates that parents participate in significantly more light-intensity activity but less MVPA and SB, compared with those without children.[28 29] For example, in the study by Gaston et al,[28] mothers took part in more light-intensity activity (p=0.001) but less MVPA (p=0.000) than women without children. Meanwhile, fathers only engaged in more light-intensity activity, compared with men without children (p=0.000). These associations differed by the number and age of children. For instance, mothers with one child, new mothers, participated in more light-intensity activity (p=0.005) than women without children, while mothers with two or three or more children took part in less MVPA (p=0.00 and p=0.005, respectively). Notably, mothers whose youngest child was aged 12–15 years engaged in more SB (p=0.017), compared with women without children, while mothers whose youngest child was aged less than 6 years participated in more light-intensity activity (p=0.000) and less MVPA (p=0.000). In fathers, those with two children took part in more light-intensity activity than men without children (p=0.000). Furthermore, fathers whose youngest child was aged less than 6 years engaged in more light-intensity activity (p=0.000) but less MVPA (p=0.000), compared with men without children. Considering that parents participate in more light-intensity activity and less MVPA and SB than those without children, it is necessary to review available evidence exploring PA by intensity and examine the relationship between PA and SB across the entirety of having a child, from the preconception period, throughout pregnancy, the postpartum period and into parenthood.

Preliminary searches broadened our knowledge of available evidence and informed the formulation of research questions. We conducted searches of Medline and Google Scholar to identify reviews exploring the impact of having a child on PA in the UK. The search terms were 'having a child' AND 'physical activity' AND 'review' AND ('United Kingdom' OR 'UK'). No relevant reviews were identified. We conducted further searches to identify relevant primary research studies by removing the search term relating to reviews. No studies were identified from Medline. Only one study was identified as relevant from the first 100 of 3080 results from Google Scholar.[30] This study by Werneck et al[30] examined the association between family-related life events (cohabitation/marriage and becoming a parent) and change in PA in the UK. Findings from this study suggest that becoming a parent was associated with a decline in PA among men (β=−0.234, 95% CI −0.396 to −0.072), but not women (β=0.126, 95% CI −0.048 to 0.301). Specifically, men who became fathers while cohabiting (β=−0.201, 95% CI −0.383 to −0.020) and without cohabiting (β=−0.937, 95% CI −1.623 to −0.250) experienced greater declines in PA compared with those remaining single and without children, where non-cohabiting fathers experienced the greatest decline. These associations did not differ by the age of children. Since no relevant reviews were identified and a dearth of primary research studies, we aim to provide an overview of available evidence exploring the impact of having a child on PA in the UK.

## METHODS AND ANALYSIS

We will conduct a review of available evidence exploring the impact of having a child on PA in the UK using best-practice scoping review methodologies. Although there is no consensus definition for scoping reviews,[31 32] it is a form of literature review which can map key concepts and available evidence within a given topic area, exploring the extent, range and nature of existing evidence.[33 34] Moreover, scoping reviews can define and clarify existing concepts, summarise and disseminate findings to stakeholders, such as practitioners and policymakers, identify knowledge gaps and determine the main priorities for future research.[33 34]

A scoping review, rather than a systematic review, is more congruent with the aims and purposes of this study. Specifically, scoping reviews use broad research questions to provide an overview of the available evidence.[33] Meanwhile, systematic reviews use focused research questions to identify the best evidence. Despite being useful as a precursor to systematic reviews, scoping reviews are legitimate in their own right as standalone studies and are not required to be followed up with a systematic review.[33] Furthermore, although scoping reviews are not, by name, systematic, they do follow a rigorous systematic process.[35 36] Unlike systematic reviews, it is important to recognise that best-practice methodological frameworks for scoping reviews do not recommend critical appraisal of individual sources of evidence, although this is still possible with reasonable justification.[35–38] Plainly, scoping reviews should not be considered a rapid and cheap alternative to systematic reviews, where they map the breadth of available literature and, as such, require a substantial amount of time and resources, including labour and finances.[33]

We will use best-practice scoping review methodological frameworks by Arksey and O'Malley,[33] adapted by Levac et al[34] and the Joanna Briggs Institute,[36 37] to guide the conduct of this review. This is an iterative process by which research questions and methods may be further developed with the emergent evidence. We will report any deviation from this protocol in the final review. In addition, we will use the Preferred Reporting Items for Systematic reviews and Meta-Analysis extension for Scoping Reviews (PRISMA-ScR) to guide the reporting of this protocol as detailed in online supplemental file 1.[39] Furthermore, methods were informed by previous scoping reviews in PA for health research.[22 40–42] This scoping review is registered with the Open Science Framework (OSF) to enhance the rigorousness and transparency of this research process.

**Patient and public involvement**
None.

## Stage 1: identifying the research questions

The following broad research questions were informed by preliminary searches and interdisciplinary discussions within the research team. Moreover, research questions were formulated using the P.C.C. (Population, Concept, Context) mnemonic which will help to identify relevant studies and establish appropriate eligibility criteria.[37]

► What is known about the impact of having a child on PA in the UK?
  – What is known about the relationship between PA and SB in this field?
► What are the knowledge gaps in this field?
► What are the main priorities for future research in this field?

## Stage 2: identifying relevant studies

### Eligibility criteria

The following a priori eligibility criteria were informed by preliminary searches, interdisciplinary discussions within the research team and research questions.

► Participants have to be adults who are reported to have been planning to have (ie, preconception, non-parents), having (ie, pregnant), have had (ie, postpartum and into parenthood, parents) or planned to have (ie, stillbirth) a child.
► Participants can be from any group, including general and specific populations (eg, those recruited based on their living with a physical or mental illness or condition).
► Participants can be of any sex, ethnicity, parity, relationship status and socioeconomic position.[43]
► Findings must explore the impact of having a child on PA, but not the impact of PA on health and well-being.
► Findings must report data on PA and/or SB which was assessed either quantitatively (ie, device-derived or through self-report) and/or qualitatively (eg, attitudes, views, perspectives, experiences, etc).
► Studies can be published literature (eg, primary research studies, including reviews) or grey literature (eg, unpublished dissertations/theses or conference proceedings).
► Studies must be of quantitative observational (eg, cross-sectional, prospective cohort), qualitative or mixed-methods study design.
► Studies must not have an experimental study design (eg, randomised controlled trials) evaluating adherence and/or acceptability of interventions, or secondary analysis thereof; guidelines, opinion pieces (eg, commentaries or editorials); or popular press articles (eg, magazine or newspaper articles).
► Studies must be conducted with data generated exclusively within the UK, including England, Northern Ireland, Scotland and Wales.
► Studies must be written in the English language.

## Search strategy

### Step 1: an initial limited search

We conducted an initial limited search of Medline (Ovid) and Google Scholar (Advanced Search) for relevant reviews. Given that there were no reviews exploring the impact of having a child on PA in the UK, we expanded search terms to allow for those using data generated elsewhere, including 'having a child' AND 'physical activity' AND 'review'. The two records identified from Medline and the first 100 of the 6720 records identified from Google Scholar were reviewed. A total of five reviews were identified as relevant.[22 25 44–46] We also conducted a search of ProQuest to identify any similar dissertations but did not retrieve any records.

### Step 2: identifying keywords and search terms

We used keywords and index terms featured within the titles and abstracts of the relevant reviews to inform main searches. Main searches will identify studies exploring the impact of having a child on PA in the UK, from the preconception period, throughout pregnancy, the postpartum period, and into parenthood, from inception to February 2022. We first prepared an a priori search strategy for published literature within the Medline electronic bibliographical database, with Boolean operators and Medical Subject Headings (ie, MeSH terms). We then adapted this search strategy for searches within CINAHL (EBSCOhost), Embase (Ovid), PsycINFO (Ovid) and Web of Science databases (Clarivate). All search strategies are detailed in online supplemental file 2. Search terms will relate to those status shifts involved in 'having a child' in an attempt to remain inclusive, where having a child can impact a variety of populations, including those of differing sex, ethnicity, parity, relationship status and socioeconomic position.[43] Moreover, search terms will relate to the main concept of 'physical activity', including 'sedentary behaviour', and the context of the 'United Kingdom'. We have made a pragmatic decision to only search for studies which have these search terms in the title and/or abstract due to the practical constraints of this review.

We will use a similar search strategy for grey literature, where the search terms used for Medline have been adapted for Google Scholar and ProQuest (online supplemental file 2). In Google Scholar, this will involve a reduced set of search terms and multiple searches due to the limitation on characters. Moreover, in Google Scholar, it is only possible to search for studies which have these search terms in the title. We will also search websites of organisations (eg, Best Beginning's, Tommy's) involved in the commissioning, undertaking or cataloguing of research in this field and, if appropriate, we will make contact by email.

### Step 3: further searching of references and citations

We will conduct backwards citation tracking of included studies to identify further potentially eligible studies. Considering that scoping reviews are typically iterative, additional search terms and information sources (eg, electronic bibliographical databases and websites) may be added to existing search strategies at any time throughout the research process.

We may also contact corresponding authors of included studies to request further information. We will make search strategies available from the OSF registration. We will import references of identified studies into EndNote V.X9.2 reference management software and organise them by database of origin, with duplicates removed.

## Stage 3: study selection

Titles and abstracts of identified studies will be predominantly screened for eligibility by one researcher (MN). A second researcher (JB) will independently complete this same process on a random sample of 10% of titles and abstracts. If the same screening decisions are taken by both researchers for 95% of studies, then MN will screen the remaining titles and abstracts. Meanwhile, if the same decisions are taken for less than 95% of studies, then MN and JB will meet to discuss these decisions with reference to the eligibility criteria and repeat this process. Any conflicts after this point will be resolved by a third researcher (CF/RP). If there is no consensus between these three parties, then we will include the study in the review. All researchers will meet to discuss the eligibility criteria before embarking on this process to minimise these occurrences.

## Stage 4: charting the data

Data charting of included studies will be predominantly conducted by MN. A data charting form will be employed, using the a priori headings detailed below.[37] Given that scoping reviews are typically iterative, additional headings may be added to the form throughout the research process. The data charting process will involve MN charting 90% of studies and JB charting 10% of studies, with JB checking 10% of MN's data and vice versa. If there are considerable differences between researchers, MN and JB will meet to discuss decisions with reference to the data charting form and rectify any issues. Any conflicts will be resolved by CF/RP during group meetings. If a full-text article cannot be retrieved, MN will attempt to gain access through the university library and by contacting corresponding authors. If the full text article is not accessible, the study will be excluded.

- ► Author(s).
- ► Year of publication.
- ► Source and country of origin.
- ► Aims/purpose.
- ► Study population and sample size (if applicable).
- ► Methodology/methods.
- ► Concept.
- ► Findings and details of these (eg, how measured) (if applicable).
- ► Key findings by theme that relate to research questions.

## Stage 5: collating, summarising and reporting the findings

We will conduct a numerical analysis of included studies to describe review decisions, findings from searches, removal of duplicates, study selection, full-text retrieval and backwards citation tracking.[37] We will present these findings using a study flow diagram. In addition, we will map data in a tabular and diagrammatic format by period of publication, country of origin, study type and themes. Considering that scoping reviews are typically an iterative process, we may include additional categories of data. We will also conduct a thematic analysis of included studies using NVivo V.12 qualitative data analysis software to provide a description of key concepts relevant to research questions and findings by theme. We will analyse these findings while being open to sociodemographic differences in the target population, such as sex, ethnicity, parity, relationship status and socioeconomic position.[43] Furthermore, we will use the PRISMA-ScR framework to guide the reporting of this study.[39]

## Critical appraisal of individual sources of evidence

We will not conduct critical appraisal of individual sources of evidence, where this is consistent with methodological frameworks for scoping reviews.[33]

## ETHICS AND DISSEMINATION

Ethical approval will not be required for this scoping review. This review will provide a novel insight into the impact of having a child on PA in the UK. Specifically, we will map key concepts and available evidence, identify knowledge gaps and determine the main priorities for future research. We aim to consult with key stakeholders, including practitioners and policymakers, to share findings and allow comment, providing greater insight with their unique expertise and perspectives.[34 47] We will subsequently summarise and submit findings for peer review and publication in a relevant open-access journal. Moreover, we will develop an infographic to be disseminated at conferences and online via professional social media channels, such as Twitter. Importantly, findings from this study will be reported regardless of the nature of the relationships between having a child and PA in the UK.

**Acknowledgements** The authors would like to thank Dr Joey Murphy for his guidance in identifying relevant literature pertaining to best-practice methodological frameworks for scoping reviews. We would also like to thank Jack Brazier (JB), a non-author, for accepting the role of second researcher within the review selection and data charting processes.

**Contributors** MN, CF, RP and FS formulated the research questions for this scoping review. CF identified the methodological frameworks and methods. MN prepared the methods. MN, CF and RP contributed to the development of the search strategy. MN wrote the first and subsequent drafts of the manuscript. All authors reviewed and approved these manuscripts.

**Funding** This study was part of MN's PhD research at the Centre for Exercise, Nutrition, and Health Sciences, School for Policy Studies, University of Bristol. MN is a PhD Researcher with the South West Doctoral Training Partnership (SWDTP) which is funded by the Economic and Social Research Council (ESRC) (grant number: ES/P000630/1).

**Competing interests** None declared.

**Patient and public involvement** Patients and/or the public were not involved in the design, or conduct, or reporting, or dissemination plans of this research.

**Patient consent for publication** Not required.

**Provenance and peer review** Not commissioned; externally peer reviewed.

peer-reviewed. Any opinions or recommendations discussed are solely those of the author(s) and are not endorsed by BMJ. BMJ disclaims all liability and responsibility arising from any reliance placed on the content. Where the content includes any translated material, BMJ does not warrant the accuracy and reliability of the translations (including but not limited to local regulations, clinical guidelines, terminology, drug names and drug dosages), and is not responsible for any error and/or omissions arising from translation and adaptation or otherwise.

**ORCID iD**
Matthew Northcote http://orcid.org/0000-0001-7563-9974

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
