## [Reviewer comments · BMJ Open]

ARTICLE DETAILS

TITLE (PROVISIONAL)	The impact of having a child on physical activity in the United Kingdom: A scoping review protocol
AUTHORS	Northcote, Matthew; Foster, Charlie; Pulsford, Richard; Spotswood, Fiona

VERSION 1 – REVIEW

REVIEWER	J Ohlendorf Marquette University College of Nursing
REVIEW RETURNED	21-Apr-2022

GENERAL COMMENTS	I look forward to seeing the results of this review, as it is an important part of public health with a population of people often overlooked--parents of young children. This protocol is also novel because very few studies or reviews address the impact of parenting on the non-birthing partner in a relationship. I have a few suggestions for improvement: 1. It is unclear to me if you intend to include parents with any aged child, or only parents of young children? What is the age span of children whose parents will meet inclusion criteria?2. Your reference for the definition of physical activity is from 1985, and there are much more recent references from which you could derive your definition.3. Because you are including an expansive definition of parenthood in relation to impact on physical activity, you may want to consider an intentional focus on whether the impact appears different based on whether the parent was the parent who gave birth or not.4. After reviewing your methods description, I'm unclear as to whether you are planning to review only articles published including only UK samples, or whether you will include articles with non-UK participants. Because you mentioned finding very few articles, I get the impression you will include articles from non-UK samples. If that's the case, then the sections where you state you will determine the impact of parenthood on physical activity in the UK is unclear.
---

REVIEWER	Jesus Ibarluzea 15. School of Psychology, University of the Basque Country UPV/EHU
REVIEW RETURNED	11-May-2022

GENERAL COMMENTS	I do not see the interest of this publication. It is so general that it could help to do this or any other scoping review.
--

REVIEWER	Samantha Fawcner University of Edinburgh, The institute for Sport Physical Education
-----------------	---

	and Health Sciences
REVIEW RETURNED	28-Jun-2022

GENERAL COMMENTS	Thank you for this opportunity to review this scoping review protocol. The review will make a significant contribution to knowledge (though see below) and the protocol proposed is suitably developed. There are some changes that you could address to strengthen the published protocol. Introduction There is a need to make a clearer case for the review. The rationale is there, but is created by a series of statements that the reader is thus expected to interpret into a need for this review. At the moment, it seems that the rationale presented in the introduction for the review is based on the following statements:- i) Having a child has been identified as being an inflection point for obesity. Please make it explicitly clear how this relates to the current study ii) An existing review that suggests that having a child reduces PA Please comment on the limitations of this existing review(s) and why the current scoping review is needed. Also, please be clear why we need a review that focuses on the UK alone? iii) That fathers and other family members may be equally impacted by this life transition Has this not been included in the previous review?? iv) that unintended births should exacerbate the impact of having a child on PA There seems limited rationale for this statement The final statement 'consequently, it is important to promote PA among those having a child in the UK' is not a derivation from the preceding introduction. Surely, you are making a case for the review (i.e understanding the impact of having a child and PA), rather than the promotion of PA to this population per se. Methods The inclusion of sedentary is important, but should not be treated as simply a minor addition (line 25, page 5). I suggest you either include Sedentary from the outset, and build into the rationale and the methods, or don't include it at all. Page 5, line 11. Participants have to be adults THAT ARE REPORTED to have been planning to have etc. otherwise this is not a helpful statement, as many participants may have had a child, but may not be a reported characteristic Page 5, line 20. Participants in inclusion are adults only, so this needs to be updated
---

	Page 7, line 23. If there is poor agreement in initial 10% then suggest that remaining titles and abstracts should be double screened. The same applies for data extraction.
--	--

VERSION 1 – AUTHOR RESPONSE

Concerning the comments of Reviewer 1, I would like to thank them for their kind interest in this scoping review and helpful suggestions for its improvement.

Firstly, it was unclear to this reviewer if we intended to include parents with any aged child, or only parents of young children. They were concerned with the age span of children whose parents would meet the inclusion criteria. This is a warranted concern where differing ages of children pose differing demands on parents and therefore their ability to participate in physical activity. We considered this exact point in the development of this protocol and, due to its formation as a scoping review, we planned to address the literature and these differences as they emerge. In this way, parents of all ages of children will be included in this study: simply including all parents. Nevertheless, to remove any ambiguity, I have included any parities for inclusion within the eligibility criteria.

Next, this reviewer commented on our choice of definition by Caspersen, Powell, and Chirstenson (1985). Although there may be more recent definitions to choose from, this one is the most common and accepted definition within this field of study. As such, this definition continues to be used within contemporary literature and even in current government documents (https://assets.publishing.service.gov.uk/government/uploads/system/uploads/attachment_data/file/832868/uk-chief-medical-officers-physical-activity-guidelines.pdf). Consequently, I will not be changing this definition.

This reviewer was further concerned with our expansive definition of parenthood in being inclusive of all peoples who experience this life event and transition. Specifically, this reviewer addressed the potential for social differentiation within studies and how we would address this within the review. In order to address this concern, I have added a statement to ‘Stage 5: Collating, summarising, and reporting the findings’, where we will inspect for social differentiation on a number of population characteristics, including sex, ethnicity, parity, relationship status, and indicators of socio-economic position.

Finally, this reviewer was confused as to whether we are planning to review only articles published including only UK samples, or whether we will include articles with non-UK participants. Moreover, since there were very few articles identified in preliminary searches of the literature, this reviewer questioned why we would not expand search terms to allow for more international studies. To clarify, we have stated that we will only include studies which generate data exclusively in the UK to provide more contextually relevant findings for this population. This means that studies must have been conducted within the UK with a UK-residing population. I feel this is clear within our eligibility criteria. Whilst there was little evidence identified from preliminary searches, we will be conducting a scoping review of published and grey literature to determine the extent of the available evidence, be that a little or a lot. Considering the reasonable rationale for this study, for which this reviewer agrees, it is inherently important to examine the range of potential literature available, hence the justification for conducting a scoping review in the first place.

Regarding Reviewer 2’s comment, I do not believe it to have been sensical, nevermind constructive to the development and subsequent publication of this protocol. As a result, I have read Reviewer 2’s comment, but it will not be acknowledged.

Upon following up as to the status of the decision for this manuscript, I was asked to provide a third, more topic-specific reviewer, for which I chose Dr Samantha Fawkner from the Physical Activity for Health Research Centre, Institute for Sport, Physical Education, & Health Sciences, University of

Edinburgh. I was aware of Dr Fawkner's research from my time studying at the University of Edinburgh and she was kind to accept my request. I chose Dr Fawkner to review this scoping review protocol where she was co-author on a similar scoping review that has informed this protocol (<https://bjsm.bmj.com/content/55/6/319.long>). Dr Fawkner is a renowned and very successful researcher within our field of physical activity for health research. I would like to thank her for these comments which have and will continue to be invaluable to the conduct of this scoping review.

I have addressed the first comments of this reviewer concerning the vagueness of our rationale by refining and expanding on all already included sections and later included sections (see below). For example, the statement on how having a child has been identified as an inflection point for obesity has been altered, where this was merely intended to represent its implications for adult health and wellbeing. In addition, the specified existing review that has cited has since been critiqued more thoroughly and paired with another more recent review in this field. I have commented on the limitations of these reviews and specified why it is important to focus on the UK alone. What's more, I removed the statement about how fathers and other family members may be impacted by this life transition, compared to mothers, where this has been considered by previous reviews. I also removed the statement that unintended births exacerbate the impact of having a child on physical activity as there is little rationale for this statement. I corrected the concluding statement of the rationale to make a case for the review rather than the promotion of PA in this population.

I agree with this reviewer that the inclusion of sedentary behaviour is important and, consequently, it will not be treated as a minor addition. As a result, I included additional paragraphs within the introduction to provide a rationale for exploring changes to sedentary behaviour relative to physical activity. Moreover, this rationale was further built into the methods. Although some studies do consider physical activity and sedentary behaviour as distinct, separate constructs, in this review, we are concerned with activities across the energy expenditure spectrum and, as such, sedentary behaviour will be considered under the 'banner' of physical activity, as in the UK Chief Medical Officers' Physical Activity Guidelines (https://assets.publishing.service.gov.uk/government/uploads/system/uploads/attachment_data/file/832868/uk-chief-medical-officers-physical-activity-guidelines.pdf).

I have amended the eligibility criteria to include adult participants 'THAT ARE REPORTED' to be having a child, given that, in many studies, participants may be having a child, but this characteristic may not have been reported. Moreover, considering that only adults were to be included in the review, I have removed the inclusion criteria for populations of any age group.

Finally, with regards to the screening and data charting processes, this reviewer suggests that if there is poor agreement in the initial 10% of screening and data charting decisions that the remaining articles should be double screened. Although this method has been featured within previous scoping reviews, we believe it is not essential within scoping review methodological frameworks (see Peters et al., 2020) and is merely one way of many to promote rigorous practice. We have chosen to screen a percentage of articles, as in previous reviews, and if the given threshold has not been met, the primary and secondary reviewer will meet to discuss decisions with reference to the eligibility criteria. The researchers will then repeat the selection/data charting process and any further conflicts will be resolved by a third reviewer. Considering that, at the time of writing, this review has largely been completed, this method allowed scrutinisation of the eligibility criteria and permitted changes to be made were there to be any. In this way, this method appears to be more congruent with scoping review methodologies which permit an iterative approach to selection and data charting processes, rather than the more rigid and laborious screening/data charting method proposed by the reviewer. Furthermore, this scoping review was part of my PhD research and, as such, I did not have the time nor resources, including labour and finances, to organise for the double screening of articles. We believe this method to be a rigorous and systematic process and, as such, suitable for publication within BMJ Open.

VERSION 2 – REVIEW

REVIEWER	J Ohlendorf Marquette University College of Nursing
REVIEW RETURNED	20-Aug-2022

GENERAL COMMENTS	Thank you for your attention to this review. You have addressed all the concerns I had and I look forward to seeing both this protocol and the eventual review in its published form. I do, still have some reservations only in the fact that you plan to include parents of children at all ages and stages in your review. Like SES, gender, etc., I do hope that this will be a point of analytical interest in the review. As a researcher and a nurse who, for 15 years, cared for families in the childbearing transition, I have seen that the childbearing and infant years have a distinct transition, the toddler years another, the school-aged years another, the adolescent parenting years another, followed by the transition out of home. Each of these brings a new developmental transition for parent, and I would hope the authors would attend to this in their presentation of findings and their discussion. That would make this review a very valuable contribution to the literature for those of us who are engaged in the work of developing interventions to promote physical activity among families navigating these transitions. Thank you again, for your work in this area, and I wish you well in your next steps.
---

REVIEWER	Samantha Fawkner University of Edinburgh, The institute for Sport Physical Education and Health Sciences
REVIEW RETURNED	20-Aug-2022

GENERAL COMMENTS	Thank you for addressing my comments thoroughly. The inclusion of SB in the protocol I see as a strength, and believe you have made a good case for this. All issues that I raised have been suitably addressed. There are some very minor typographical/syntax errors so suggest careful proof reading and the next stage. I look forward to reading the review
---

VERSION 2 – AUTHOR RESPONSE

Next, we would like to thank Reviewer 1, Dr. J Ohlendorf from Marquette University College of Nursing for their continued efforts with this manuscript – your comments are invaluable. This reviewer remained concerned ‘only in the fact that [we] plan to include parents of children at all ages and stages in [our] review’. To address this concern, I want to reassure this reviewer that this will be a point of analytical interest in the review when examining socio-demographic differences in the target population. We plan on reviewing articles which explore the impact of the life transition of having a child, from the preconception period, throughout pregnancy, the postpartum period, and into parenthood, on physical activity in the United Kingdom. In this way, we plan to review all eligible studies which focus on physical activity across these life stages and so the ages of the most recent child-to-be are implied throughout this process. Nevertheless, we acknowledged that the parental status, the parity, of the target population is distinctly important and so we will determine whether

participants within studies were primiparous or multiparous in this regard. In this way, we will be able to discern the differences between those in the target population who are to be new parents and those who are more established parents, with existing children of differing ages. Although the number and ages of existing children is important, we are aiming to examine physical activity, as our outcome, across the relevant life transition of having a child, which can happen to both new and established parents. Given that this is a scoping review, we will do our best, in the stated ways, to address these differences within this study.

Finally, we would like to thank Reviewer 3, Dr. Samantha Fawcner from the University of Edinburgh – your topical and methodological insight has gone a long way to making this protocol potentially publishable within this journal. In this reviewer's words, 'all issues that [they] raised have been suitably addressed'. Despite this, this reviewer notes 'some very minor typographical/syntax errors' and suggests 'careful proof reading and the next stage'. I have noted these typographical/syntax errors whilst conducting these minor revisions and further proof readings. These can be viewed within the 'Main document – marked copy'.